# COVID-19 Infection Percentage Estimation from Computed Tomography Scans: Results and Insights from the International Per-COVID-19 Challenge

**DOI:** 10.3390/s24051557

**Published:** 2024-02-28

**Authors:** Fares Bougourzi, Cosimo Distante, Fadi Dornaika, Abdelmalik Taleb-Ahmed, Abdenour Hadid, Suman Chaudhary, Wanting Yang, Yan Qiang, Talha Anwar, Mihaela Elena Breaban, Chih-Chung Hsu, Shen-Chieh Tai, Shao-Ning Chen, Davide Tricarico, Hafiza Ayesha Hoor Chaudhry, Attilio Fiandrotti, Marco Grangetto, Maria Ausilia Napoli Spatafora, Alessandro Ortis, Sebastiano Battiato

**Affiliations:** 1Institute of Applied Sciences and Intelligent Systems, National Research Council of Italy, 73100 Lecce, Italy; fares.bougourzi@u-pec.fr; 2Laboratoire LISSI, University Paris-Est Creteil, Vitry sur Seine, 94400 Paris, France; 3Department of Computer Science and Artificial Intelligence, University of the Basque Country UPV/EHU, Manuel Lardizabal, 1, 20018 San Sebastian, Spain; fadi.dornaika@ehu.eus; 4IKERBASQUE, Basque Foundation for Science, 48011 Bilbao, Spain; 5Institut d’Electronique de Microélectronique et de Nanotechnologie (IEMN), UMR 8520, Universite Polytechnique Hauts-de-France, Université de Lille, CNRS, 59313 Valenciennes, France; abdelmalik.taleb-ahmed@uphf.fr; 6Sorbonne Center for Artificial Intelligence, Sorbonne University of Abu Dhabi, Abu Dhabi P.O. Box 38044, United Arab Emirates; 7College of Computer Science and Technology, Taiyuan University of Technology, Taiyuan 030024, China; chaudharysuman560@gmail.com (S.C.);; 8School of Computing, National University of Computer and Emerging Sciences, Islamabad 44000, Pakistan; 9Faculty of Computer Science, Alexandru Ioan Cuza University, 700506 Iasi, Romania; 10Institute of Data Science, National Cheng Kung University, No. 1, University Rd., East Dist., Tainan City 701, Taiwan; 11Dipartimento di Informatica, Universita degli Studi di Torino, Corso Svizzera 185, 10149 Torino, Italy; davide.tricarico@aitemsolutions.com (D.T.); hafizaayeshahoor.chaudhry@unito.it (H.A.H.C.);; 12Department of Mathematics and Computer Science, University of Catania, 95125 Catania, Italybattiato@dmi.unict.it (S.B.)

**Keywords:** COVID-19, convolutional neural network, deep learning, segmentation, Per-COVID-19, transformer, estimation

## Abstract

COVID-19 analysis from medical imaging is an important task that has been intensively studied in the last years due to the spread of the COVID-19 pandemic. In fact, medical imaging has often been used as a complementary or main tool to recognize the infected persons. On the other hand, medical imaging has the ability to provide more details about COVID-19 infection, including its severity and spread, which makes it possible to evaluate the infection and follow-up the patient’s state. CT scans are the most informative tool for COVID-19 infection, where the evaluation of COVID-19 infection is usually performed through infection segmentation. However, segmentation is a tedious task that requires much effort and time from expert radiologists. To deal with this limitation, an efficient framework for estimating COVID-19 infection as a regression task is proposed. The goal of the Per-COVID-19 challenge is to test the efficiency of modern deep learning methods on COVID-19 infection percentage estimation (CIPE) from CT scans. Participants had to develop an efficient deep learning approach that can learn from noisy data. In addition, participants had to cope with many challenges, including those related to COVID-19 infection complexity and crossdataset scenarios. This paper provides an overview of the COVID-19 infection percentage estimation challenge (Per-COVID-19) held at MIA-COVID-2022. Details of the competition data, challenges, and evaluation metrics are presented. The best performing approaches and their results are described and discussed.

## 1. Introduction

During the COVID-19 pandemic, medical imaging was widely used as a complementary or main test for COVID-19 diagnosis [1,2]. In fact, medical imaging has shown its ability to complement or even replace the reverse transcription polymerase chain reaction (RT-PCR) test, which has a considerable false negative rate [1]. In addition, RT-PCR is expensive, laboratory- and time-consuming, and not always available, especially in less developed countries [1,2,3,4]. On the other hand, medical imaging can not only be used to detect the infected persons, but it can also provide more details about the severity and progression of the infections [3,4,5,6].

To evaluate COVID-19 infection from the medical imaging, an expert radiologist is required [7]. In fact, during the COVID-19 pandemic, healthcare resources were overwhelmed, and medical staff, including radiologists and doctors, were depleted [8,9]. With the spread of the COVID-19 pandemic and the huge number of infected persons, it is very important to provide efficient artificial intelligence solutions that can aid medical staff in the fighting against future pandemics [10]. In the last years, many efficient approaches have been proposed to detect, diagnose, and evaluate COVID-19 infection using various medical imaging modalities [11,12]

In general, the quantification of COVID-19 infection from CT scans is performed by segmenting the infected regions from the CT slice, which is then compared with the entire lung regions on the same CT slice [13,14]. However, creating adequate segmentation datasets to train deep learning methods is a tedious task, which was especially true during the pandemic [9,15,16]. In this paper, a new framework is investigated to quantify, evaluate, and follow up COVID-19 infection using CT scans. This framework is based on considering the quantification of COVID-19 infection as a regression task.

The aim of the Per-COVID-19 [17] challenge is to evaluate the effectiveness of the recent deep leaning methods to learn from noisy data for a sophisticated regression task. To this end, participants were provided with training data along with the ground truth for the COVID-19 infection percentage (CIP) ground truth. Participating teams developed their approaches using this training data and optimized them using validation data via the CodaLab platform [18]. Both the training and validation CT data were estimated by two radiologists. In the final phase, the best teams on the validation data tested their approaches on the testing data, which were created using three segmentation datasets that provided a more accurate COVID-19 infection percentage. The final rankings are determined by considering both the validation and testing results, thereby addressing a variety of challenges ranging from learning and optimizing using noisy data to crossdataset evaluation.

In this paper, the Per-COVID-19 challenge is presented, which was proposed and organized as part of the 1st International Workshop on Medical Imaging Analysis for COVID-19 (MIA-COVID-2022), which was held in conjunction with the ICIAP 2022 Conference. Six teams’ approaches are described, and their results are summarized and discussed. The participating teams investigated and developed a variety of solutions, including different deep learning methods (such as CNNs and Transformers), loss functions, end-to-end training and embedding feature space learning, single and ensemble estimators, transfer learning strategies, and data augmentation techniques. The obtained results showed promising results and prove the efficiency of the proposed regressing framework for COVID-19 infection quantification as a regression task. In addition to accurate estimation, this framework presents a promising solution in a pandemic situation, as it requires less time and effort from radiologists in the labeling process. Furthermore, the results are easy to be stored and reviewed for following up the infection evolution and the response to the treatments. This framework can help in studying and predicting the behavior of this disease.

The remainder of this paper is organised as follows: Section 2 presents a comparison between the segmentation and regression scenarios for CIPE. In Section 3, the Per-COVID-19 challenge framework is described. Section 4 summarises the challenge phases. In Section 5, the best six teams approaches are presented. Section 6 summarises and analyses the results of the best six teams. The results of the Per-COVID-19 challenge are discussed in Section 7. Finally, Section 8 concludes this paper.

## 2. Regression vs. Segmentation

The aim of this section is to compare between CNN-based segmentation architectures and CNN-based regression architectures for estimating COVID-19 infection using CT scans. To this end, three segmentation datasets were used to create training and test splits for the CIPE, which are summarized in Table 1. Initially, the three datasets were divided into 75% and 25% portions. Subsequently, the union of the three 75% portions was utilized as training data, while the union of the three 25% portions served as the test data. For the CNN-based segmentation architectures, well-known segmentation architectures were used, namely Unet [19], Att-Unet [20] and Unet++ [21]. In this scenario, two models were trained for COVID-19 infection segmentation and lung segmentation, respectively. The CIPE was inferred using the infection and lung masks as follows:(1)$$CIPE=100\times \frac{\sum Infection\;pixels}{\sum Lungs\;pixels}$$

In the regression scenario, two powerful CNN architectures (ResNext-50 and DenseNet-161) were used to estimate the COVID-19 infection percentage directly from the CT slices. Two loss functions were also tested. The first and second loss functions are the standard MSE loss and the dynamic Huber loss [22]. Table 2 summarises the obtained results using three evaluation metrics, which are the mean absolute error (MAE), root mean square error (RMSE), and Pearson correlation coefficient (PC).

Based on these results, we can conclude that the regression architectures outperformed the segmentation architectures on all evaluation metrics (MAE, PC, and RMSE). These results show that estimating the CIPE as a regression task is more effective than the segmentation approach. On the other hand, the results show that using the dynamic Huber loss function improved the performance for both Resnext-50 [23] and DenseNet-161 architectures [24], with the DenseNet-161 performing the best with the dynamic Huber loss. We believe that the regression architectures achieve a better CIPE than the segmentation architectures because the automatic segmentation of infected regions is a more complicated task that requires more training data to achieve higher performance. However, creating a large segmentation dataset is time-consuming and labour-intensive, which is difficult to achieve in a pandemic situation.

**Table 1 sensors-24-01557-t001:** The summary of the segmentation datasets used.

Name	Dataset	#CT-Scans	#Slices
Dataset_1	COVID-19 CT segmentation [25]	40	100
Dataset_2	Segmentation dataset nr. 2 [25]	9	829
Dataset_3	COVID-19-CT-Seg dataset [14]	20	3520

**Table 2 sensors-24-01557-t002:** The comparison between Segmentation and Regression approaches for COVID-19 infection percentage estimation.

Architecture	MAE	PC	RMSE
Unet	5.2	0.8931	10.62
AttUnet	4.98	0.8292	12.02
Unet++	4.94	0.8509	11.30
ResNext-50 (MSE)	3.15	0.9653	5.85
ResNext-50 (Huber)	2.65	0.9696	5.31
DenseNet-161 (MSE)	3.18	0.9688	5.48
DenseNet-161 (Huber)	2.72	0.9718	5.14

## 3. Challenge Framework

### 3.1. Data Preparation

The Per-COVID-19 challenge dataset consists of three splits: Train, Val, and Test. The Train and Val splits were collected from two hospitals between June 2020 and February 2021, as shown in Table 3. COVID-19 infection was confirmed by RT-PCR and two radiologists, who confirmed the presence of COVID-19 infection manifestations in the CT scans. Each patient has only one CT scan in this dataset, and the CT scans consist of 40–200 slices. A total of 189 CT scans from 189 patients were used to create these two sets. Table 4, summarises the Train and Val splits. The training set was derived from 132 CT scans, wherein 128 CT scans were confirmed to have COVID-19, and 4 CT scans showed no signs of infection (Healthy). The validation set was obtained from 57 CT scans, with 55 CT scans confirmed to have COVID-19 infection, and 2 CT scans showing no signs of infection (Healthy).

In order to minimize the effort needed from the radiologist in the labelling phase because of the pandemic situation and to obtain as much diversity as possible in patient age, sex, morphology, and level and type of infection, the Covid 19 percentage was estimated from several slices from each CT scan. Radiologists intentionally avoided considering slices with a similar infection distribution and type. Additionally, infected slices from the upper, middle, and bottom sections of the lung were chosen [17]. It is worth noting that the estimation was carried out directly by radiologists, thereby inferring the percentage of infected areas to lung areas. While performed by experienced radiologists, human estimation inherently introduces noise into the procedure.

On the other hand, the testing set is the union of the three COVID-19 segmentation datasets that were used in Section Table 2. These datasets are the following: COVID-19 CT segmentation [25], Segmentation dataset nr.2 [25], and COVID-19-CT-Seg dataset [14], as summarised in Table 5. For the test data, the ground-truth of the CIPE was obtained by calculating the ratio of the infected pixels over the lungs pixels. In fact, the ground truth percentages obtained from the segmentation datasets are more accurate than the estimated ones by the radiologists. However, segmenting the infection requires huge effort, and it is very time-consuming [9,15,16].

### 3.2. The Competition Challenges

To create efficient machine learning tools, adequate and well-labelled data are required. During the COVID-19 pandemic, creating such a dataset was a very difficult task due to the fact that the radiologists and doctors had been exhausted. To deal with these limitations, semisupervised approaches have been widely investigated [15,16,26]. Other works proposed to learn from noisy data such as [9]. This competition was designed to follow the last path, where the radiologists were asked to estimate the infection without segmenting both infection and lung regions, which is a tedious task that requires a lot of effort and time.

In addition to the difficulty of learning from noisy data, COVID-19 infection percentage estimation is a very challenging task, since the infection has a high variability in shape, location, type, and intensity. Other factors that makes the estimation of the infection from the CT scans more challenging include the patient age and gender diversity, the infection severity gradations, and the CT scan slices’ morphological changes from the first, middle, and last slices.

Figure 1 shows some examples from the Train, Val, and Test splits. As mentioned in Section 3.1, the Train and Val splits were contracted from two centres. On the other hand, the Testing split was created from three segmentation datasets, which were collected from different centres, scanners, and recording settings. As shown in Figure 1, the Testing split has variety of imaging illumination and contrast unlike the Train and Validation splits. This work aims to study the generalization ability of the competition approaches with respect to the real scenario, where each CT scan appears differently according to the scanner and the recording setting.

### 3.3. Model Training

The participants to the challenge were asked to estimate the percentage of COVID-19 infections for each slice using machine learning. Only ImageNet’s pretrained models and lung nodules segmentation models were allowed to be used. In addition, using external data or other pretrained models were not allowed in order to guarantee that all participants had the same training and testing data and the same data resources accessibility.

**Figure 1 sensors-24-01557-f001:**
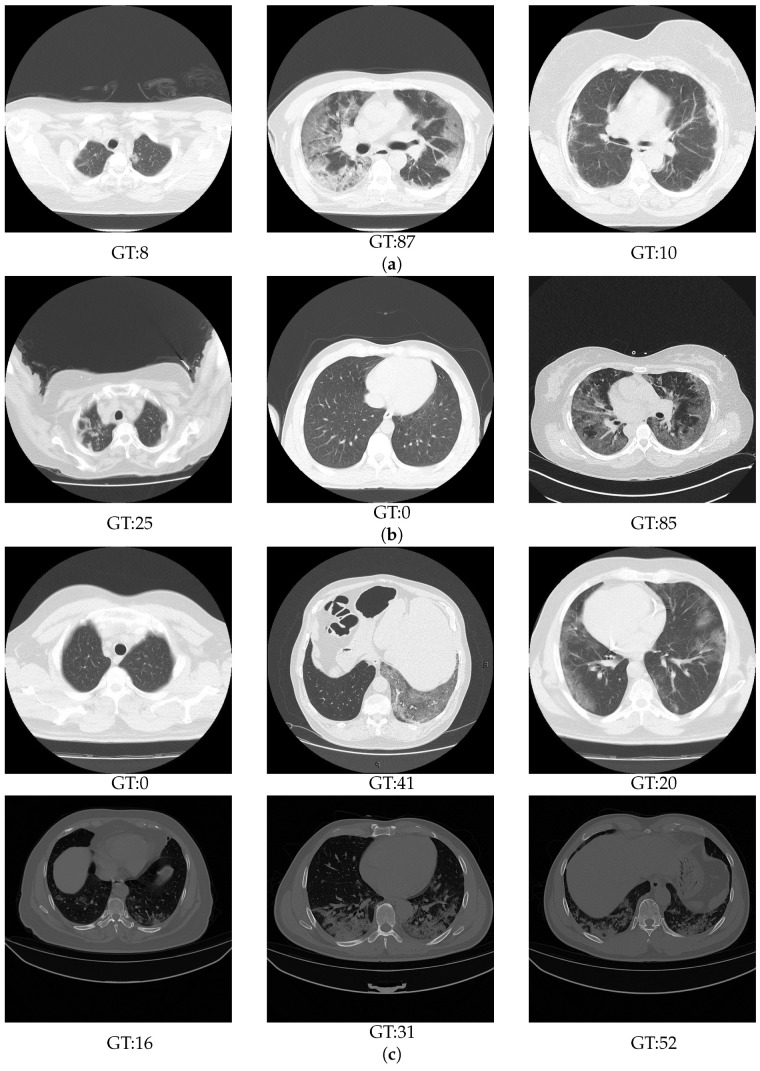
Some examples of the challenge splits samples: (**a**) Train, (**b**) Validation, and (**c**) Test.

### 3.4. Model Evaluation

To evaluate the performance of the baseline and challenge teams’ approaches, three evaluation metrics were used: mean absolute error (*MAE*), root mean square error (*RMSE*), and Pearson correlation coefficient (*PC*), which are defined in Equations (Equation 2)–(Equation 4), respectively [27].
(2)$$MAE=\frac{1}{n}\sum_{i=1}^{n}|y_{i}-\hat{y_{i}}|$$
(3)$$RMSE=\frac{1}{n}\sum_{i=1}^{n}{\left(y_{i}-\hat{y_{i}}\right)}^{2}$$
(4)PC=∑i=1n(yi−yi¯)(yi^−yi^¯)(∑i=1n(yi−yi¯)2∑i=1n(yi^−yi^¯)2
where $Y=(y_{1},y_{2},\ldots ,y_{n})$ are the ground truth COVID-19 percentages of the testing data, which consist of *n* slices, and $\hat{Y}=\left(\hat{y_{1}},\hat{y_{2}},\ldots ,\hat{y_{n}}\right)$ are their corresponding estimated percentages. For Equation (Equation 4), $\bar{y_{i}}$ and $\bar{\hat{y_{i}}}$ are the means of the ground truth percentages and the estimated ones, respectively.

The MAE and RMSE are error indicators, with smaller values indicating better performance. On the other hand, the PC is a statistical measure of the linear correlation between two entries *Y* and $\hat{Y}$. A value of 1 means that there is a completely positive linear correlation, and 0 means that there is no linear correlation.

In determining the final ranking, both validation and testing results were taken into consideration, as each set presented its own unique challenges. Additionally, the labels for the validation data were not disclosed to all challenge teams. The evaluation was conducted on the CodaLab platform, which allowed for only a limited number of submissions per day. The final evaluation scores are calculated as follows:(5)$$MAE_{final}=0.3\times MAE_{val}+0.7\times MAE_{test}$$
(6)$$RMSE_{final}=0.3\times RMSE_{val}+0.7\times RMSE_{test}$$
(7)$$PC_{final}=0.3\times PC_{val}+0.7\times PC_{test}$$
where $MAE_{val}$, $RMSE_{val}$, and $PC_{val}$ are the obtained results on the validation data, and $MAE_{test}$, $RMSE_{test}$, and $PC_{test}$ are the obtained results on the testing data. For the final ranking, $MAE_{final}$ is considered as the most important evaluation criterion. In the case of two or more competitors achieving the same $MAE_{final}$, then the $PC_{final}$ and the $RMSE_{final}$ are considered as the tiebreaker metrics, respectively.

### 3.5. Baseline Method

As a baseline method, the Densnet-161 [24] architecture with a dynamic Huber loss function was considered, since it achieved the best performance in Section 2. To have a good learning ability, a model pretrained on ImageNet [28] was used. Densnet-161 was trained for 30 epochs, and the batch size was set to 20 images. The initial learning rate was set to $1\times 10^{-4}$, which decays by 0.1 after 10 epochs two times. For the dynamic Huber loss function, ${\mathrm{betta}}_{\max}$ and ${\mathrm{betta}}_{\min}$ were set to 15 and 1, respectively. To reduce overfitting, the input image was rotated using a random angle between $-10^{\circ }$ to $+10^{\circ }$ to serve as active data augmentation. The baseline result was obtained by training Densnet-161 on the training data, where the best model on the validation data that corresponded to the best MAE was used to predict theCIPE on the testing data of the challenge.

## 4. Challenge Phases

The Per-COVID-19 challenge details and results are available on Github (https://github.com/faresbougourzi/COVID-19-Infection-Percentage-Estimation-Challenge) (accessed on 27 February 2024).

### 4.1. Validation Phase

The training and validation phase was carried out on the CodaLab platform. For the validation set, each team had the opportunity to send ten predictions per day in CodaLab. At the end of the validation phase, the teams that performed better than the baseline had the possibility to participate in the testing phase.

### 4.2. Testing Phase

In the testing phase, the top teams that had achieved the best results in the validation phase were allowed to pass to the testing phase.

### 4.3. After the Challenge Ended

After the end of the challenge timeline, the competition was reopened for the evaluation on the validation data to allow new teams to evaluate the performance of their approaches. The validation evaluation results are accessible on the CodaLab platform through the following links: https://competitions.CodaLab.org/competitions/35575 and https://CodaLab.lisn.upsaclay.fr/competitions/7065 (accessed on 27 February 2024). The team that attained the best results on the validation data was granted the opportunity to assess the performance of their approach on the testing data.

## 5. Participating Teams

By the end of the validation phase, more than 80 teams had registered for the competition on the CodaLab platform, and more than 60 teams submitted their entries through the CodaLab platform. Of these, 15 teams performed better than the baseline result. In the end, the best six teams participated in the MIA-COVID-2022 workshop (https://sites.google.com/view/covid19iciap2022/home (accessed on 27 February 2024)). In this section, the best six approaches are described.

### 5.1. Taiyuan_university_lab713

The Taiyuan_university_lab713 team from the University of Taiyuan, China [29] proposed to use a Swin-L transformer [30] as a feature extractor. To have a good feature extraction capability, the pretrained weights on ImageNet [28] were used. After extracting the features, the multilayer perceptron (MLP) architecture, comprising two layers, was trained on these features to estimate the COVID-19 infection. To identify the optimal model for the validation data, the training data were partitioned into 10 folds, and crossvalidation was conducted to test various training splits. Additionally, an ensemble of the ten models on the validation data was examined.

### 5.2. TAC

The TAC team [31] proposed the SEnsembleNet architecture for the COVID-19 percentage estimation. The SEnsembleNet approach was developed to leverage the learning capabilities of six Resnet variants through a squeeze and excitation (SE) block [32]. Specifically, the SEnsembleNet approach consists of two stages. First, six Resnet variants (Resnest50d [33], Resnet-RS-50 [34], SeresNext-50_32x4d [32], Ecaresnet50t [35], SkresNext-50_32x4d [36], and SeresNet-50 [32]) featuring modification of their FC layers by adding two MLP layers were trained independently. In the second stage, the MLP layers were removed, and each CNN architecture was used as a feature extractor by freezing its layers. The deep features of the six trained models were stacked, then an SE block was added to weight the features of the backbone CNNs. Finally, a decision FC layer was attached to the SE block to obtain COVID-19 percentage estimation. It should be noted that only the SE block was trained in the second phase.

### 5.3. SenticLab.UAIC

The SenticLab.UAIC team from the University of Alexandru Ioan Cuza, Iasi, Romania [37] proposed a multilosses approach with CNN backbones. For this purpose, two FC layers were added to the CNN backbone. The first FC layer consists of 101 neurons corresponding to discrete percentage scores (0–100), and the second FC layer consists of one neuron for the CIPE. In addition, three loss functions were considered: (i) a Smooth-$L_{1}$ loss function between the predicted COVID-19 infection percentage and the ground truth (GT), (ii) for the second loss, a Smooth-$L_{1}$ loss was calculated between the expected prediction (sum of the first FC layer’s output probabilities) and the GT, and (iii) a KL divergence loss was calculated between the softmax of the output of the first FC layer and the probability distributions of the ground truth. The overall loss is the sum of the three losses. It should be noted that only the CIPE output was considered in the test phase. For the backbone CNN, a modified version of ResNeSt [33] was introduced, where the GAP was replaced by a hybrid pooling mechanism. In addition, the ensemble of a few trained models of ResNeSt improved the results compared to using a single model.

### 5.4. ACVLab

The ACVLab team from the University of National Cheng Kung, Taiwan [38] proposed a hybrid Swin transformer [30] architecture exploiting the multitask strategy. In more details, their approach consists of three main components: (i) a maximum rectangle approach was used to detect the lung regions; (ii) a two-head hybrid Swin transformer was introduced, in which the first head is used for regression losses and the second one for classification loss; and (iii) the overall loss is the summation of two regression losses ($L_{1}$ and $L_{2}$) and classification loss. In this last one, the infection score range was divided into levels, and crossentropy loss was applied between the classification heads and the GT level.

### 5.5. EIDOSlab_Unito

The EIDOSlab_Unito team from the University of Turin, Italy [39] proposed a contrastive learning-based approach for the CIPE. Their approach consists of four main steps: (i) the input slices were preprocessed using image resizing and pixel intensity scaling to reduce the brightness and pixel intensity contrast between the training and test splits; (ii) DenseNet-121 [24] architecture was used as a feature projector to the 1024-dimensional target space; (iii) a distance loss function was proposed to correlate the distances between the samples in the deep feature space and their GT values in the target space (i.e., the COVID-19 level of infection); and (iv) the COVID-19 percentage estimation of a new image, the query, is computed by averaging the values from the nearest neighbours by means of Euclidean distance in the projection feature space from images in the training set (the reference set).

### 5.6. IPLab

The IPLab team from the University of Catania, Italy [40] proposed a mix-up data augmentation technique to create virtual examples that significantly increase the diversity of the training data for the CIPE. In their approach, two images with their GT COVID-19 infection percentages were mixed by using a random parameter α that ranges between 0 and 1. In addition to the mix-up data augmentation, other data augmentation methods were exploited such as Gaussian blurring, color jittering, and random horizontal and vertical flipping. As a backbone architecture, the pretrained Inception-v3 [41] on ImageNet [28] was used.

## 6. Challenge Results

### 6.1. Results

Table 6, Table 7 and Table 8 show the validation, test, and final results. The results on the validation data (Table 6) show that SenticLab.UAIC achieved the best performance and improved the results compared to the baseline by 1.07, 1.26, and 0.0165 for the MAE, RMSE, and PC, respectively. On the other hand, the difference in performance between the first and second approach is small on the three evaluation metrics. Similarly, the second team outperformed the third one only on the MAE by a tiny difference of 0.02. The third team outperformed the second one for both the PC and RMSE metrics. In the same way, Teams 4, 5, and 6 achieved close results, where Team 4 outperformed the other two teams on the MAE metrics, and Team 5 outperformed the other two teams on both the PC and RMSE. From these results, it can be seen that all teams achieved comparable results, although both the Train and Val ground truths had some labelling noise.

From Table 7, it is noticed that the proposed methods by the teams show high generalization ability compared to the baseline. For instance, the best team outperformed the baseline by 5 and 5.11 for the MAE and RMSE, respectively, and by 0.2203 for the PC metric. On the other hand, we notice that Taiyuan_university_lab713 and TAC performed the best in the test phase and rank first and second, respectively. These two teams outperformed the other teams in the three evaluation metrics by a significant margin. Teams 3, 4, and 5 achieved close results, with team SenticLab.UAIC achieving the best MAE, and team EIDOSlab_Unito achieving the best PC and RMSE results. Although the IPLab team achieved the worst MAE, it achieved close results to the fourth team based on the PC and RMSE metrics.

Based on the validation and test result, Table 8 summarises the final ranking. From this, it can be seen that the Taiyuan_university_lab713 and TAC teams achieved the best performance and show high generalization ability in both validation and test splits. In the following, Teams 1 to 6 correspond to the final raking order (rank 1 to 6), respectively.

### 6.2. Analysis

Table 9 summarises the key components of the top six teams. The following elements are considered in this table: preprocessing, deep learning backbones and architectures, loss functions, deep features, pretrained weights, ensembling, and data augmentations. In summary, preprocessing strategies were not widely explored, with only two teams that investigated preprocessing and only one team that attempted to extract lung regions. As a backbone, all teams used deep learning approaches, where four teams used CNN-based approaches, and the remaining two teams used transformer-based approaches [42]. It is worth noting that two of the top four teams used transformer-based methods, thereby proving that transformers provide a very efficient solution for medical imaging tasks. In terms of the loss function, several losses were investigated by each team, starting from the standard $L_{1}$ and MSE loss functions to the Smooth-$L_{1}$ and Huber loss functions. In addition to the standard losses, the multitask loss function [43] was explored by combining regression and classification losses as proposed by the ACVLab and SenticLab.UAIC teams. The EIDOSlab.Unito team proposed a contrastive learning approach [44] that combines regression and Euclidean distance loss functions.

Deep features were utilized in three approaches, including the two best-performing teams in the challenge. This underscores the efficiency of employing deep learning approaches as feature extractors and subsequently incorporating a regression block. Pretrained weights on ImageNet were employed by five teams, thereby highlighting the significance of leveraging transfer learning for enhanced feature extraction capabilities. Notably, two out of the three top-performing teams utilized ensemble approaches. TAC’s approach involved an attention-based ensemble approach on top of six CNN models’ deep features. In contrast, SenticLab.UAIC’s approach exploited the simple averaging of deep features from five trained hybrid ResNeSt models. Additionally, data augmentation techniques were widely employed and investigated by all six teams to augment the training data and mitigate overfitting.

## 7. Discussion

Figure 2, Figure 3 and Figure 4 show a comparison between the COVID-19 infection percentage GT and the six teams’ predictions from three 3D CT scan examples from the test data. Figure 2, Figure 3 and Figure 4 represent the minimal, extentsive, and severe cases, respectively. From Figure 2, it is noticed that the predictions of Teams 1, 2, 4, and 5 match the ground truth in most of the upper, middle, and lower lung slices. In contrast, the predictions of Teams 3 and 6 are about 2% greater than the GT in most slices.

From Figure 3, we notice that almost all of the prediction curves match the GT one. More precisely, the second and the fifth approaches predicted the first two slices incorrectly by a large margin. In contrast, the other teams’ predictions are close to GT. In the last slices, most of the approaches decreased to 0% before the ground truth. Similar remarks are observed in the third example (Figure 4), where most of the six approaches failed to match the GT in the beginning and the end of the CT scans. It is likely that these two regions were not well covered in the training data, as the infection appears clearly in the middle slices and can be very confusing at the beginning and end of the CT scans.

Figure 5 shows eight test examples with the corresponding GT and the predictions of the six models. The first example shows that most of the models had a CIPE close to the ground truth, with the worst prediction being the prediction of Team 5, which achieved an absolute error (AE) of 6%. On the other hand, Teams 1 and 2 had the best performances, whose AE values were 1% and 2%, respectively. Also, the second and the third examples show that most of the teams achieved a CIPE close to the GT, especially the first three teams. To discover the most difficult aspects for the proposed approaches, the remaining five examples were selected based on the most confusing slices for the teams’ approaches. The fourth example is a severe slice (CIP = 50–75%). In this example, it is noticeable that the fourth team achieved the best performance (AE = 6%), while the fifth team overestimated the infection (AE = 26%). In the fifth test example (another severe slice), all teams overestimated the infection percentage, where the third team achieved the closest CIPE with an AE equal to 12%, which is a relatively large error. The last three examples are more challenging examples, where the illumination and the contrast are very different from the training as shown in Figure 1. It is noticed from the sixth and seventh examples that the second and the third approaches have the best generalization ability compared with the other approaches. On the other hand, the fifth and the sixth teams were the worst, with the fifth team overestimating the infection and the sixth team failing to recognize the infected regions. The last example is a lower lung slice, where only a small portion of the lung appears; it was very confusing to most of the challenge’s approaches as shown in Figure 4.

In this section, it is noticed that most of the challenge approaches face two main challenges. The first one is that the top and bottom slices of the 3D CT scans are very confusing. This issue could be solved by augmenting the labelled slices from these regions. The second major challenge is the acquisition of the CT scans using different scanners and settings. This issue could be solved by even adding more CT scans from other centres to the training dataset or by using effective data augmentation techniques that deal with the variation of the contrast and illumination. Furthermore, using lung segmentation can help to deal with these challenges, since the region of interest will be predefined.

## 8. Conclusions

The Per-COVID-19 challenge provided benchmark methods and results for estimating the COVID-19 infection percentage (CIPE) from CT scans. In this challenge, new methodologies were proposed and investigated, thereby addressing various challenges such as training with noisy data and evaluation in multicentre, multivendor, and multirecording settings. Over 175 teams participated in this challenge, with 24 teams achieving results superior to the baseline. This paper summarizes and discusses the approaches and results of the top-six teams.

The developed approaches demonstrated promising performance in learning from noisy data and exhibited good generalization ability in crossdataset scenarios. These results offer valuable insights for monitoring infection progress. The current study can provide an efficient tool for studying and predicting the behaviour of the disease with limited efforts and resources, particularly in pandemic situations where swift action can save many lives.

## 9. Limitations

The authors would like to state that this technology is intended not to substitute physicians or radiologists, but it can be a valid tool for better diagnosis provided by experts. The reported works should not be used to overdiagnose COVID-19 by substitution of an expert medical team by a partially validated artificial intelligence system.

## Figures and Tables

**Figure 2 sensors-24-01557-f002:**
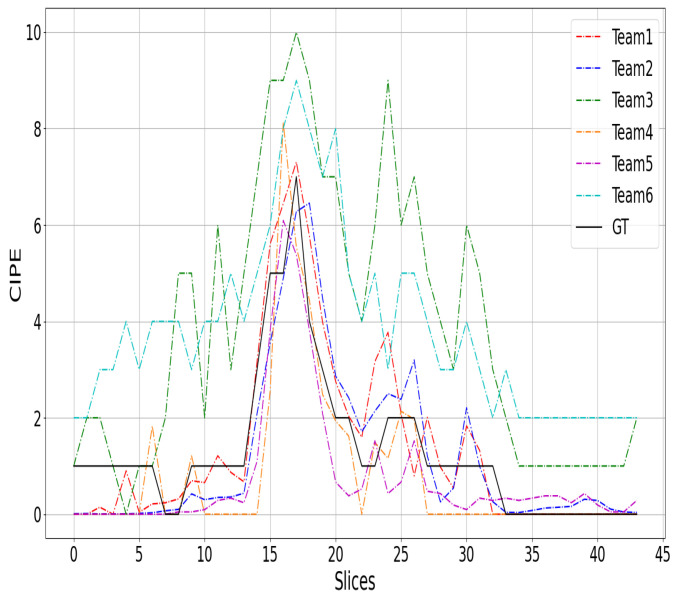
The top-six teams’ predictions (rank of Table 8) and the GT curves for a 3D CT scan of minimal cases.

**Figure 3 sensors-24-01557-f003:**
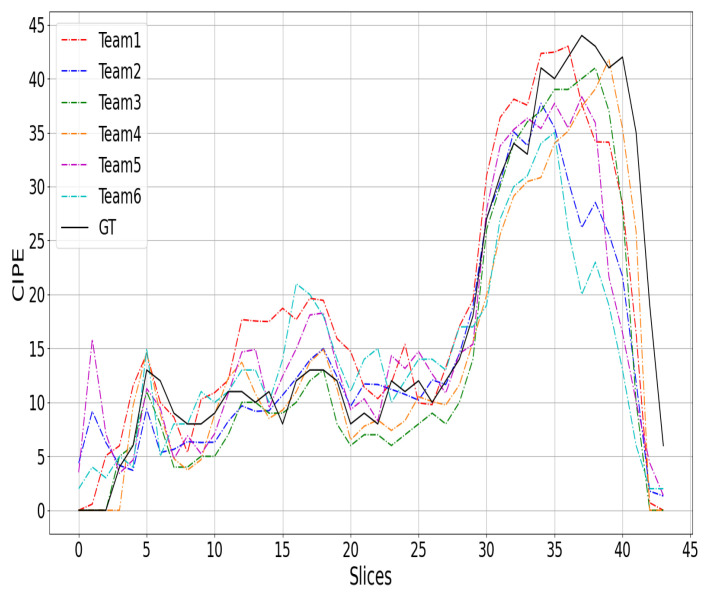
The top-six teams’ predictions and the GT curves for a 3D CT scan of extentsive cases.

**Figure 4 sensors-24-01557-f004:**
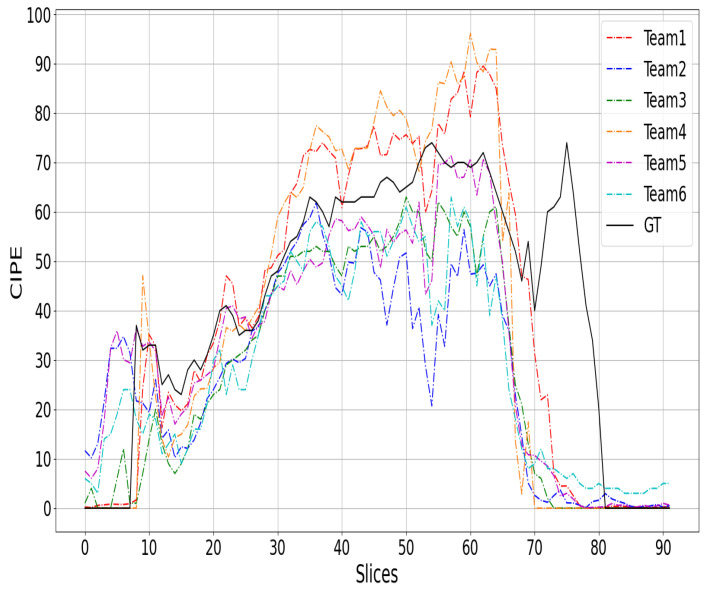
The top-six teams’ predictions and the GT curves for a 3D CT scan of severe cases.

**Figure 5 sensors-24-01557-f005:**
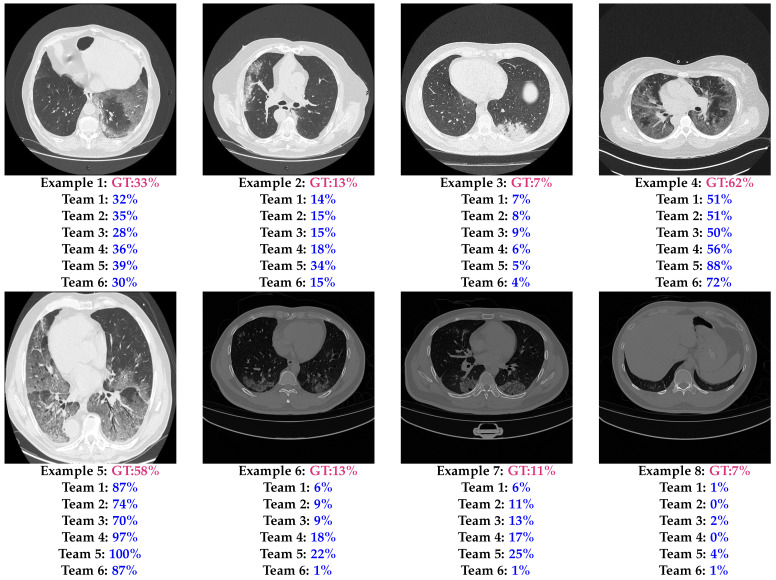
COVID-19 infection percentage estimation examples of the six best teams and the corresponding ground truth on the testing data.

**Table 3 sensors-24-01557-t003:** The collected Per-COVID-19 dataset construction for the Train and Val splits.

Hospital	Hakim Saidane Biskra	Ziouch Mohamed Tolga
#CT-scans	154	35
Device Scanner	Hitachi ECLOS CT Scanner	Toshiba Alexion CT Scanner
Slice Thickness	5 mm	3 mm

**Table 4 sensors-24-01557-t004:** Train and Val splits of Per-COVID-19 challenge.

Split	#CT Scans	#Slices
Train	132	3054
Val	57	1301

**Table 5 sensors-24-01557-t005:** Per-COVID-19 testing dataset summary.

Name	Dataset	#CT-Scans	#Slices
Dataset_1	COVID-19 CT segmentation [25]	40	100
Dataset_2	Segmentation dataset nr. 2 [25]	9	829
Dataset_3	COVID-19-CT-Seg dataset [14]	20	3520
Test Set	Total	69	4449

**Table 6 sensors-24-01557-t006:** Per-COVID-19 validation results.

Rank	Team	MAE	PC	RMSE
1	SenticLab.UAIC	4.17	0.9487	8.19
2	TAC	4.48	0.9460	8.54
3	Taiyuan_university_lab713	4.50	0.9490	8.09
4	EIDOSlab_Unito	4.91	0.9429	8.70
5	ausilianapoli94	4.95	0.9435	8.60
6	ACVLab	4.99	0.9364	9.08
-	Baseline	5.24	0.9322	9.45

**Table 7 sensors-24-01557-t007:** Per-COVID-19 testing results.

Rank	Team	MAE	PC	RMSE
1	Taiyuan_university_lab713	3.55	0.8547	7.51
2	TAC	3.64	0.8022	8.57
3	SenticLab.UAIC	4.61	0.7634	9.09
4	ACVLab	4.86	0.7287	10.27
5	EIDOSlab_Unito	5.02	0.7977	9.01
6	IPLab	6.53	0.7091	9.97
-	Baseline	8.57	0.6344	12.62

**Table 8 sensors-24-01557-t008:** Per-COVID-19 final ranking.

Rank	Team	MAE	PC	RMSE
1	Taiyuan_university_lab713	3.84	0.8830	7.92
2	TAC	3.89	0.8453	8.55
3	SenticLab.UAIC	4.48	0.8190	8.46
4	ACVLab	4.90	0.7910	9.43
5	EIDOSlab_Unito	4.98	0.8413	8.79
6	IPLab	6.06	0.7794	9.01
-	Baseline	7.57	0.7237	11.66

**Table 9 sensors-24-01557-t009:** The challenge approaches summary.

Team	Preprocessing	Backbone	Architecture	Loss Function	Deep Features	Pretraining	Ensemble	Data Augmentation
1. Taiyuan _university _lab713	None	Transformer	Swin-L MLP	MSE	✓	ImageNet	✗	Hue Saturation Brightness Contrast
2. TAC	None	CNN	ResNest-50d ResNetrs-50 SeresNext-50 EcaresNet-50t Skresnext-50 Seresnet-50 SEnsemble-Net	Smooth-$L_{1}$	✓	ImageNet	✓	Horizontal flipping Shift scale rotation
3. SenticLab.UAIC	None	CNN	ResNeSt-50 with Hybrid Pooling	Smooth-$L_{1}$ Distribution loss KL-divergence	✗	ImageNet	✓	Rotation Color Jittering Contrast Brightness Sharpness ShearX -ShearY Cutout TranslateX TranslateY
4. ACVLab	Maximum- Rectangle Extraction	Trans-former	Hybrid Swin	$L_{1}$ MSE Cross-Entropy	✗	None	✗	Horizontal flipping Random shifting Random scaling Rotation Hue Saturation Brightness Contrast adjustment
5. EIDOSlab _Unito	Pixel intensity scaling	CNN	DenseNet-121 Contrastive learning	$L_{1}$ Euclidean distance	✓	ImageNet	✗	Horizontal flipping -Random Cropping
6. IPLab	None	CNN	Inception-v3	Huber	✗	ImageNet	✗	Mix-up Gaussian blurring Color jittering Vertical flipping

## Data Availability

The data were collected from the Hospitals of Hakim Saidane Biskra and Ziouch Mohamed Tolga (Algeria), and they were made available at https://github.com/faresbougourzi/Per-COVID-19 (accessed on 4 August 2023).

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
