# Peer review of "COVID-19 Infection Percentage Estimation from Computed Tomography Scans: Results and Insights from the International Per-COVID-19 Challenge"

_sensors, 2024, doi:10.3390/s24051557_

Round 1

Reviewer 1 Report

Comments and Suggestions for Authors

The author explained and compared the technical solutions of the participating teams. The article is organized reasonably.

However, in '6.2 Analysis' part, the author just described a few techniques that might be useful for enhancing performance. More analysis can be provided with regard to the issues and data characteristics that make these technical solutions appropriate for this competition.

Author Response

Thank you for this comment. Adding more analysis for the proposed solution requires doing further experiments and investi- gations. We aim to cover these aspects in our future works.

Reviewer 2 Report

Comments and Suggestions for Authors

the study is well written ,however if the authors add sensity and accuracy of top six method in table ,it will enhance the data validation.

In adition ,the manuscript is too long .it need shorten text.

Author Response

Thanks for this comment. Due to the fact that the challenge is a regression task, it is impossible to provide the sensitivity and accuracy. On the other hand, we believe that all of the paper sections are necessary to the challenge summary.

Reviewer 3 Report

Comments and Suggestions for Authors

This paper can be improved by correcting several typos, adding missing commas,  taking care of acronyms throughout the paper and reformulate some sentences. 

I have highlighted all the positions in the attached version of the paper were I consider that some changes are necessary. I will further discuss only the most important ones.

CodaLab should appear like this all over the paper.

The sentence at rows 51-52 need to be reformulated.

MIA-COVID-2022 should appear like this all over the paper.

Several sections of the paper use the Present Tense, while the competition took place in 2021-2021. Please take care and use the Past Tense when appropriate. 

Check out all the designations for deep learning backbones. Like Denesenet-161 on row 85. But it applies on ResNext and ResNeSt also.

Please reformulate the sentence at rows 121-122.

Please reformulate the sentence at rows 175-177 and explain here daily limited submissions, as is done latter in the paper.

I think that the sentence at rows 191-193 should be a little bit reformulated to become clearer. The word baseline is very important for the discussion over the results.

Please reformulate the sentence at rows 209-210.

Maybe the sentence at rows 220-221 should contain more specific information about the inputs and outputs of the MLP classifier.

All over the paper the articles “aan or the” are missing before singular nouns. In English they are mandatory. For instance at row 227, “proposed the SEnsembleNet” would be correct.

dimensional at row 268

mix-up at row 275, 279

I don’t think we can say “ground truths are noisy” on row 293. Please reformulate.

In the paragraph on rows 324-328, you state that only 3 teams used deep features. The next line says the 5 teams used pretrained weights on ImageNet. These are equivalent to deep features, and hence the contradiction!

The name of Table 9 contains a sentence that could be found in text also. I think it is not necessary here.

Also in Table 9 there is the first mentioning of using Hue Saturation and Color Jittering. My problem is that CT-scans are not color photos, they have only 256 levels of gray, so they have no Hue or Color. Please clarify this with the teams using them.

In the caption of Figure 5 it is not necessary to capitalize the words like in a paper title.

Comments on the Quality of English Language

It should be improved. No author is English and that can explain some mistakes.
